REGISTERED REPORT PROTOCOL

# Distrust within protected area and natural resource management: A systematic review protocol

**Brian D. Erickson⊙\*, Kelly Biedenweg**

Department of Fisheries, Wildlife, and Conservation Sciences, Oregon State University, Corvallis, Oregon, United States of America

\* brian.erickson@oregonstate.edu

**Data Availability Statement:** The data generated from this review is posted to our existing pre-

## Abstract

Trust is a key variable for successful natural resource management and is commonly the focus of conceptual and methodological development. Distrust, on the other hand, is frequently cited as an obstacle to management, but appears to be rarely defined, conceptually underdeveloped, and inconsistently examined. This systematic review protocol (OSF pre-registration https://doi.org/10.17605/OSF.IO/GKUAW) was developed to answer two primary questions in relation to protected area and natural resource management: 1) How is distrust conceptualized, and 2) What methods are used to gather evidence of distrust? The aim is to provide a comprehensive overview of how distrust is theoretically developed and what questions are used to uncover distrust. Also, it will summarize findings on what leads to and results from distrust. Four academic and eight gray literature databases will be searched using Boolean keyword searches. Articles eligible for inclusion are those that present original research, gather and present evidence of distrust, and focus on protected areas and/or natural resource management. The review will result in a narrative synthesis that summarizes approaches to distrust within protected area and natural resource management.

## Introduction

### Rationale

Trust is a key variable for successful natural resource management. When defined, it is most commonly taken to be "a psychological state comprising the intention to accept vulnerability based upon positive expectations of the intentions or behaviors of another" [1]. Alternatively, trust is "confident positive expectations regarding another's conduct," where confident positive expectations entails "a belief in, a propensity to attribute virtuous intentions to, and a willingness to act on the basis of another's conduct" [2]. A third less common definition of trust is "choosing to risk making something you value vulnerable to another person's actions" [3]. These definitions emphasize a willingness to be vulnerable and expectations of another's behavior; however, many measures of trust focus on one aspect or the other, not both [4]. Trust is linked to increased approval of management decisions, minimized resistance to

registration on OSF, which is cited in the abstract (https://doi.org/10.17605/OSF.IO/GKUAW).

**Funding:** Publication of this paper was supported, in part, by the Thomas G. Scott Publication Fund. No additional external funding was received for this study.

**Competing interests:** The authors have declared that no competing interests exist.

planning efforts and protected areas, improved message reception, and decreased management costs [5, 6]. It is also essential for public engagement, collaboration, and decision making (e.g., [5, 6]).

In contrast, distrust is often recognized as a major obstacle to effective natural resource management, leading to fear, skepticism, opposition, and rancor [5, 7, 8]. Although distrust appears to be rarely defined or theoretically developed, it is thought to both drive and result from resource conflict [9]. It is said to undermine constructive debate and public inquiry and pose a major obstacle to finding common ground, compromising, and collectively solving problems [9, 10]. However, distrust can also motivate engagement in public decision making [8]. Concepts of "critical trust" [11] and "effective distrust" have been proposed to explain a healthy middle ground of trust and distrust, where stakeholders "trust but verify" [2]. According to Poortinga and Pidgeon [11] "critical trust" occurs when both general trust and skepticism are high and leads to a situation where "one may be willing to rely on information, but one is still somewhat skeptical, and thus may still (constructively) question the correctness of the received information." "Effective" distrust is a similar point where "questioning of institutions in this sense is not destructive, but can be seen as an essential component of political accountability in a participatory democracy" [12].

While laypeople recognize a practical difference between trust and distrust [8], social scientific research appears inconsistent in its conceptualization and methods for gathering evidence of distrust [5, 13, 14]. Some scientists approach distrust as equivalent to low/no trust, others treat trust and distrust as opposites, and still others see distrust as a distinct concept from trust. As such, it has been difficult to determine the antecedents (determinants) and outcomes of distrust. We will systematically review the conceptualization and methods for gathering evidence of distrust in the protected area and natural resource management literature.

## Theoretical background

**Conceptualizations of trust in natural resource management.** Within natural resource management literature, there is disagreement over the dimensionality of trust. Some authors argue that trust is multidimensional. These scholars treat the processes underlying trust as complex and based on substantial information, including ability, benevolence, integrity, competence, sincerity, credibility, consistency, inclusiveness, caring, fairness, openness, responsibility, reliability, equity, and perceived risk and benefits (e.g., [7, 15]). Other scholars argue that trust is unidimensional. They claim that although various subcomponents may influence trust judgements, these subcomponents can be difficult to separate from each other, especially in regard to public trust in government agencies [6, 16–20]. Still other scholars argue that trust is a bidimensional construct. For example, Parkins [21, 22] uses Poortinga & Pidgeon's [11] typology of trust, which identifies "general trust" (incorporating competence, care, fairness, and openness) and "skepticism" (involving issues of credibility, bias, and vested interest).

Another area of debate regards whether, in what contexts, and to what degree trust judgments are cognitive- or affect-based. Cognitive-based (a.k.a. "calculus-based") trust is seen as rational, calculative, and based on considering risk and benefits from available information, data, and experience [8]. In contrast, affect-based (a.k.a. "identification-based") trust "emerges from shared values, common goals, emotional bonds, identification with others' interests, concerns, and intentions; and judgments of other parties' value- systems, ethical and moral character" [8].

Stern & Coleman [5], in a synthesis of trust theory in natural resource management, identify four "types" of trust: dispositional, rational, affinitive, and procedural. Dispositional trust is "the general tendency or predisposition of an individual to trust or distrust another entity in

a particular context." Rational trust is similar to cognitive-based trust. Affinitive trust is similar to affect-based trust. Procedural trust is "trust in procedures or other systems that decrease vulnerability of the potential trustor, enabling action in the absence of other forms of trust." In contrast, some scholars consider procedures to be a referent (the thing being trusted) instead of a "type" of trust (e.g., [8]).

Scholars also identify and focus on different levels of trust. Most often, trust is researched at interpersonal, organizational, or social levels. Interpersonal trust is seen as trust between two individuals and is based on expectations grounded in personal experience [23]. Organizational trust is trust within (intra-) or between (inter-) organizations [24–26]. Social trust carries multiple meanings inside and outside of natural resource management literature. While it can sometimes refer to a generalized trust in others (dispositional trust), social trust in natural resource management contexts often refers to "the willingness to rely on those who have the responsibility for making decisions and taking actions related to the management of technology, the environment, medicine, or other realms of public health and safety "[20]. Notably, discussions of trust "levels" are different from measurements of "levels of trust," the latter referring to how much trust one party has in another party.

In summary, there are varying conceptualizations and approaches to studying trust, including dimensionality, types, and levels. Despite the lack of convergence, trust is seen as an important area of research because of its implications for natural resource management processes and outcomes.

**Conceptualizations of distrust broadly.** Typically, distrust is not defined when the term is used in the literature. When distrust is defined, it is often framed in reciprocal terms to trust. For example, distrust has been defined as "a psychological state, comprising the unwillingness to accept vulnerability, based on pervasive negative perceptions and expectations of. . . motives, intentions, or behaviors" [27]. Lewicki et al. [2] defined distrust as "confident negative expectations regarding another's conduct." Confident negative expectations include "a fear of, a propensity to attribute sinister intentions to, and a desire to buffer oneself from the effects of another's conduct." A third approach defined distrust as "the belief that a person's values or motives will lead them to approach all situations in an unacceptable way" [23]. These authors elaborated that "distrust is engendered when an individual or group is perceived as not sharing key cultural values." In contrast to the literature definitions, the Oxford English dictionary defines the verb "distrust" in two ways, "1.a. To have a doubt or dread of; to suspect. 1.b. to be without confidence in. 2.a To do the opposite of trusting; to withhold trust or confidence from; to put no trust in, or reliance on, the statements or evidence of. 2.b. To entertain doubts concerning; to call in question the reality, validity, or genuineness of; not to rely upon." These varying definitions illustrate that distrust has been conceptualized in multiple ways. The presence of competing conceptualizations suggests that distrust is an ambiguous term. When undefined, researchers may not be reporting on the same construct.

Historically, distrust was ignored across disciplines as a topic of research [27]. As a result, distrust, mistrust, low trust, lack of trust, absence of trust, and untrustworthiness were used interchangeably [27]. When mentioned, distrust was treated as the low end of a single continuum (e.g., [28, 29]) (Fig 1A).

Other researchers treat trust and distrust as two ends of the same continuum (Fig 1B). This approach implies a middle state where people neither trust nor distrust the party in question [13] and distinguishes between low trust and distrust. Bertsou [13] suggested this was an improvement in conceptualization, but argued that it failed to account for differences in emotions and behaviors associated with trust and distrust. Additionally, scholars suggest treating trust and distrust as opposites may not incorporate the fact that reasons for trust can be disconfirmed but reasons for distrust cannot easily be proven unjustified [13].

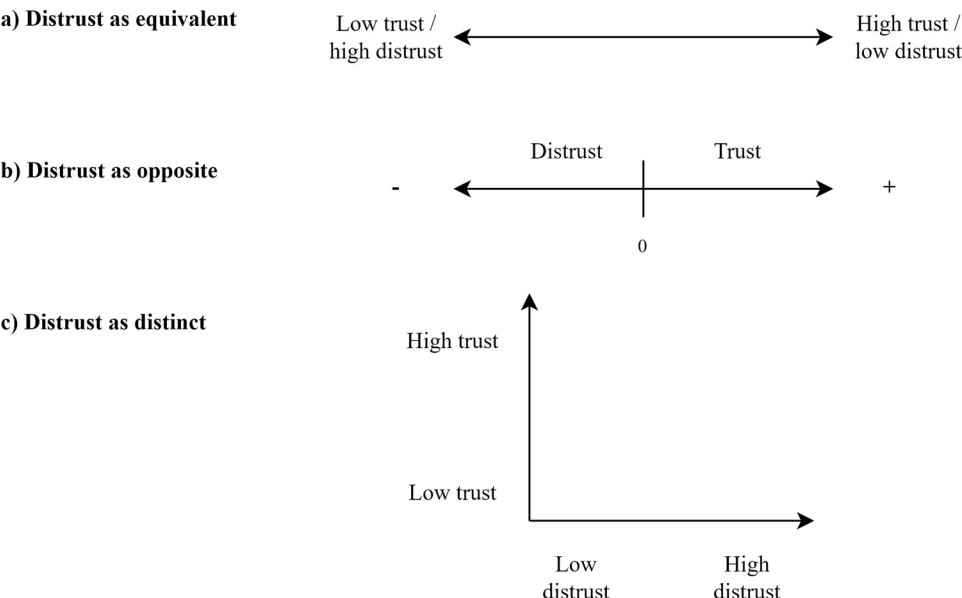

**Fig 1. Three conceptualizations of distrust in relation to trust.** (a) Low trust as equivalent to distrust. (b) Trust and distrust as opposites along the same continuum. (c) Trust and distrust as distinct and separate concepts (modified from [13]).

A third conceptualization argues that trust and distrust are separate and distinct, albeit linked, concepts [2, 13, 27] (Fig 1C). While this approach sometimes defines trust and distrust in reciprocal terms (e.g., confident positive expectations versus confident negative expectations), it explicitly argues that trust and distrust are not on a single continuum. As Lewicki et al. [2] emphasized, "low distrust is not the same thing as high trust, and high distrust is not the same thing as low trust." Proponents of this view argue that the determinants (antecedents) and effects (outcomes) differ between trust and distrust [2, 27], which reinforces their argument that trust and distrust are not opposites. In addition to theoretical arguments in favor of treating trust and distrust as distinct [2, 27, 30], empirical evidence–mostly from business, organizational and health care settings–is beginning to support this conceptualization as well (e.g., [31–39]).

Within the "distrust as distinct" conceptualization, there are two competing views [27]. One sees distrust as pervasive and, when present, it prevents the possibility of trust [23, 36]. From this perspective, distrust is theorized as a self-reinforcing cyclical process that pervades across domains [27, 40]. Distrust is conceived as "an all-encompassing negative lens through which distrusted others are perceived" [27]. Once there is distrust, all actions become suspect, and motivations and values are questioned regardless of the other party's intentions. The other "distrust as distinct" conceptualization views trust and distrust as distinct but co-existing [2, 36], with different combinations of trust and distrust each ranging from low to high. Despite both conceptualizations viewing trust and distrust as distinct, they also make differing claims about whether trust and distrust can co-exist. Part of this disagreement may be due to the authors' use of differing definitions of distrust.

In summary, additional disagreements exist regarding distrust. While distrust is often implicitly assumed to be synonymous with low trust, at least two other conceptualizations are possible. With competing conceptualizations, results discussing "distrust" may be less comparable than they appear on the surface.

**Conceptualizations of distrust in natural resource management.** Within natural resource management contexts, distrust is regularly mentioned, albeit much less often than trust. However, based on a preliminary literature search, it seems that distrust is rarely defined. For example, Davenport et al. [7] and Smith et al. [15] are regularly cited natural resource management papers (124 and 73 citations, respectively; Web of Science, January 2022) that emphasized distrust within the background and results, but neither paper defined distrust. Other papers define and explain trust and then use the term "distrust" without definition (e.g., [18, 41]). One of the few explicit definitions of distrust in natural resource management litera-ture that we have found comes from Stern & Coleman, who wrote

[distrust]. . . is conceptually distinct from a mere lack of trust. While a lack of trust indicates the absence of a specific judgment about trust, distrust refers to a state in which the trustor (entity a) believes that the trustee (entity b) will perform an action that will actually be harmful to the trustor. Distrust implies an active misgiving of entity b on the part of entity a. We describe in the following section how the antecedents of distrust may fall within the same theoretical categories as the antecedents of trust. In any case, trust may exist on a con-tinuum, ranging from complete distrust through a lack of trust toward complete trust [5].

In this passage, Stern & Coleman argue against the assumption that trust and distrust are equivalent (Fig 1A), and instead advocate for trust and distrust as opposites on a single contin-uum (Fig 1B). This is in contrast to a separate paper by Stern & Baird [14], who introduced a two "spectra" approach, citing Lewicki et al.'s [2] distrust as distinct conceptualization (Fig 1C).

Trust researchers in natural resource management usually employ the "distrust as equiva-lent" conceptualization. When studying trust through surveys, many researchers ask respon-dents to what extent they agree or disagree with statements about trust in a management agency (e.g., [16]). Disagreement is treated as low trust or distrust. Some examples of "distrust as opposite" appear in the literature. An often cited example comes from Cvetkovich & Winter [42], who asked residents, "To what extent do you trust the [Forest Service] in their fire man-agement efforts?" Response options ranged from "completely distrust" to "completely trust." Notably this same manuscript measures trustworthiness, often treated as a synonym of trust even though they are theoretically distinct [43], on a scale from "not trustworthy" to "completely trustworthy." At least one example of the "distrust as distinct" conceptualization appears in natural resource management contexts [44]. However, such treatments appear rare.

## Research questions

Questions for this review were developed after noting a discrepancy in the literature between more commonly included conceptualization and emphasis on measurement of "trust" and the limited theorization, definition, and intentional gathering of evidence of "distrust." We say "intentional" because multiple studies we encountered during preliminary literature searches set out to examine management challenges, broadly (i.e., they do not specifically set a goal of examining distrust) and then distrust emerged as a major barrier (e.g., [45, 46]).

**Primary question.** In relation to protected area and natural resource management,

- How is distrust conceptualized?

- What methods are used to gather evidence of distrust?

   **Secondary questions.**

- How, if at all, is distrust defined?

- When distrust emerges during research, how, if at all, is it conceptualized?

- How aligned are conceptualizations of distrust with the methods used to gather evidence of distrust?

- What antecedents (determinants) lead to distrust?

- What are the outcomes of distrust?

- What actions are recommended to reduce distrust?

## Methods

### Search strategy

Our search strategy was developed based on systematic review best practices [47–49] and in consultation with a university research librarian. After developing a list of potential academic and gray literature databases and search keywords, we tested our search strategies. Due to challenges filtering and exporting search results, two gray literature databases were removed. The final search strategy will include four academic and eight gray literature databases (Table 1). These databases will be searched using the keywords for distrust, natural resource management, and protected areas shown in Table 2. Database-specific Boolean search strategies were developed and follow the general format: distrust terms AND (natural resource management terms OR protected areas terms) (S1 File).

Table 1. Databases included in the systematic review.

| Database | URL |
| --- | --- |
| **Academic literature** | |
| Web of Science | www.webofknowledge.com |
| ProQuest | www.proquest.com |
| Aquatic Sciences and Fisheries Abstracts | |
| Agriculture and Environmental Sciences Collection | |
| EBSCOhost | www.ebsco.com/products/research-databases |
| Agricola | |
| Fish, Fisheries, & Aquatic Biodiversity Worldwide | |
| GreenFILE | |
| Wildlife & Ecology Studies Worldwide | |
| PsycInfo (APA Psych Net) | psycnet.apa.org |
| **Gray literature** | |
| CGIAR | www.cgiar.org/research/publications |
| Center for International Forestry Research | www.cifor.org/knowledge/publications |
| National Oceanic and Atmospheric Administration: Institutional Repository | repository.library.noaa.gov |
| National Park Service: Integrated Resource Management Applications | irma.nps.gov/DataStore/Search |
| UN Food and Agricultural Organization[a] | www.fao.org/documents/search |
| USAID: Development Experience Clearinghouse[a] | dec.usaid.gov |
| USDA Forest Service: Treesearch | www.treesearch.fs.fed.us |
| US Fish and Wildlife Service: National Digital Library | digitalmedia.fws.gov/digital/search |
| World Bank | openknowledge.worldbank.org |
| WorldFish | www.worldfishcenter.org/publications/search |

[a]Database tested during exploratory search but will be excluded in the systematic review.

**Table 2. Search keywords.**

| Distrust terms | Protected area terms |
|---|---|
| distrust* | "protected area*" |
| mistrust* | "protected forest*" |
| untrust*cynic* | "protected landscape*" |
| skeptic* | "protected seascape*" |
| sceptic* | "special protection area*" |
| "no* trust*" | "special area* of conservation" |
| "lack of trust" | "conservation area*" |
| "absence of trust" | "wilderness area*" |
| | "management area*" |
| **Natural resource management terms** | "national heritage area*" |
| "natural resource management" | "biodiversity area*" |
| "fisher* management" | "bird area*" |
| "wildlife management" | "locally managed marine area*" |
| "rangeland* management" | "indigenous and community conserved area*" |
| "forest management" | "nature reserve*" |
| "land management" | "marine reserve*" |
| "catchment management" | "forest reserve*" |
| "freshwater management" | "reserved forest*" |
| "coastal management" | "biosphere reserve*" |
| "marine management" | "conservation reserve*" |
| "ecosystem-based management" | "community reserve*" |
| "ecosystem management" | "private reserve*" |
| "environmental management" | "national park*" |
| "collaborative management" | "natural monument*" |
| "co-management" | "national heritage place*" |
| "adaptive management" | "wildlife sanctuar*" |
| "spatial management" | "national estuarine research reserve*" |
| | "communal forest*" |
| | "conservation zone*" |
| | "spatial closure" |
| | "spatial plan*" |

Articles eligible for inclusion are those that gather and present evidence of distrust, relate to protected area and/or natural resource management, and present original research. No study design, publication year, participant, intervention, comparator, or outcome limits will be imposed on the search.

## Article screening

Search results will be uploaded into Rayyan QCRI [50], an internet-based software program that facilitates collaboration among reviewers during the study screening process. The lead author (BE) will screen the titles, abstracts, keywords, and publication names for all articles identified through the search procedure. KB will go through the same process with a subsample of 10% of the original search results. To determine which studies to sample, studies will be ordered alphabetically, and KB will start with the first study and will evaluate every 10th study in the list. Each article will be coded "include," "exclude," or "unsure.". All articles meeting study topic inclusion criteria and including any of the distrust keywords will be marked as include and passed to the next round of screening.

Coding disagreements will be discussed using negotiated agreement, and revisions to criteria may be made if needed [51]. Intercoder reliability will be reported using Krippendorff's alpha with a target α ≥ 0.65. Additionally, intercoder agreement–the percent of coding discrepancies that were resolved through discussion [51]–will also be reported.

Articles that are coded as "include" by at least one reviewer or "unsure" by both reviewers will go through to full-text screening, where they will be assessed for eligibility based on the full content of the article. BE will review all articles and KB will review every 10[th] article. Agreement will be assessed using the same criteria from round 1. If K-alpha < 0.65, we will conduct negotiated agreement, clarify inclusion criteria, and then complete a third round of screening using the same procedure as in round 2, except KB will screen new articles. During full-text screening, BE and KB will determine whether articles mentioning distrust keywords are gathering and interpreting evidence of "distrust" or just low/lack of trust (see eligible evidence of distrust)

Neither of the review authors will be blind to information about the studies being reviewed (titles, journal, study authors, or authors' institutions). For each round of screening, reviewers will be blind to each other's inclusion/exclusion decisions until both have completed the screening process. To ensure literature saturation, we will manually search the reference lists of all articles that meet this study's inclusion criteria (see below) and will screen these additional publications for inclusion using the procedures already described. BE will use a standardized Excel form to extract data from each study that is determined to meet the eligibility criteria. This data extraction form will be piloted with a small subset of studies and modified before applying to the rest of the studies to be reviewed.

## Eligibility criteria

**Eligible study topics.** Any setting related to protected areas or natural resource management. The IUCN defines a protected area as "a clearly defined geographical space, recognized, dedicated and managed, through legal or other effective means, to achieve the long term conservation of nature with associated ecosystem services and cultural values" [52]. They identify six categories of protected areas: strict nature reserve, wilderness area, national park, natural monument or feature, habitat/species management area, protected landscape or seascape, and protected areas with sustainable use of natural resources. The search will also include the management of natural resources. We will use the journal Society & Natural Resources' broad definition of natural resources as including "water, air, wildlife, fisheries, forests, natural lands, urban ecosystems, and intensively managed lands." It can be difficult to determine what counts as "natural resource management" and definitions vary. The keywords and publication name (e.g., Journal of Environmental Management, Society & Natural Resources) will be used as additional guides to determining whether this is related to protected area or natural resource management.

**Eligible evidence of distrust.** Articles must gather and present evidence of distrust between one party and another party or a control system. Parties can be individuals, groups, or organizations. Control systems are "procedures, rules, contracts or other monitoring mechanisms that guide behavior" [53]. Trust in control systems is sometimes referred to as procedural trust. In this review, evidence of distrust can be gathered using quantitative or qualitative approaches.

Articles that include the phrase "low trust," "lack of trust," "absence of trust," "no trust," "not trust-," or "untrust-"must make a clear interpretation of these results as distrust. For example, Winter & Cvetkovich [54] included an item with an eight-point scale ranging from "do not trust the Forest Service at all" to "trust the Forest Service completely." They divided

this scale into those who trust (rating between 5–8) and those who distrust (1–4 rating). Thus, they measured trust, but they clearly interpreted low trust as "distrust." This article would be included in this review. In contrast, Shindler et al. [55] measure trust and confidence in the Forest Service, and mention distrust in their discussion, but do not clearly interpret their findings as indicating distrust. It is implied through the discussion, but not stated. This article would be excluded. Similarly, Abbas et al. [56] measured "lack of trust," but only mentioned "distrust" once and never mentioned "mistrust" or other synonyms. Thus, it is unclear whether "lack of trust" was a measure of trust or if it indicated distrust. The authors do not clearly interpret it as the latter. To assess the risk of under-representation of articles of this type, we will track articles excluded for not clearly interpreting trust findings as distrust.

**Eligible study types.** Articles must be original research that presents data collected by the author(s). Theoretical, review, and opinion/perspective articles will not be included.

## Study validity assessment

This review will generate a narrative synthesis of existing approaches to researching and conceptualizing distrust within protected area and natural resource management. All studies meeting inclusion criteria, including those that are not peer-reviewed and regardless of their risk of bias, will be included in the synthesis. Whether a study has been peer-reviewed will be noted in the data extraction spreadsheet.

## Data extraction

Information on each of the studies will be collected in an Excel spreadsheet made available to all study authors. The spreadsheet will provide places to record the following information, if available, for each article under review. To ensure the study is repeatable, BE will extract data from all articles and KB will extract data from every 10th article. Agreement will be assessed using the same criteria used for article screening. If K-alpha < 0.65, we will conduct negotiated agreement, clarify data extraction approaches, and then complete an additional round of extraction using the same procedure as before, except KB will screen new articles.

**Meta-data extraction.**

- Name of study reviewer (options: BE, KB)

- Full study citation

- Peer-reviewed? (Yes, No)

- Continent(s) where the study is based

- Country(ies) where the study is based

- Method type for gathering evidence of distrust (code as qualitative, quantitative, mixed)

- Sample size (n =?)

- Sample population (who was sampled?)

- Comments on methods

**Data extraction.**

- Trust definition

- Distrust definition

- Distrust conceptualization (code as equivalent, opposite, distinct, or unable to determine)

- Sample text using trust & distrust (record example text from the article showing how "trust" and "distrust" are used)

- Distrustor (who is doing the distrusting?)

- Distrustee (who is being distrusted?)

- How were the questions developed? (code as Adapted, New, Replicated, Unclear)

- Source of adapted/replicated questions (include citation)

- Number of questions gathering evidence of distrust

- Questions asked to gather evidence of distrust

- Response scale(s) used, including anchors

- Comments on questions used

- Index reliability statistics reported

- Construct validity reported

- Comments on reliability and validity

- Reported evidence of distrust (copy and paste article text)

- Reported associations with distrust (what factors were found to be related to distrust?)

- Reported antecedents of distrust (what factors were found to lead to distrust?)

- Reported outcomes of distrust (what factors were found to result from distrust?)

- Comments on distrust-related results

- Study recommendations for trust building and/or distrust reduction

- Additional comments

## Data synthesis

No quantitative, statistical synthesis of data (meta-analysis) will be conducted. Thematic coding will be used to identify and interpret patterns in the data relevant to the research questions. A systematic narrative synthesis will be provided with the information presented in text and tables to summarize and explain the characteristics and findings of the included studies. All studies meeting inclusion criteria, regardless of their risk of bias, will be included in the synthesis.

The following is a tentative outline of how we will synthesize findings. First, distrust conceptualizations will be summarized. This will include a table of any definitions provided by distrust researchers and a bar graph of the frequency of equivalent, opposite, distinct, and unclear conceptualization. Second, the antecedents of distrust will be summarized. This will likely include grouping related concepts and may include a comparison to proposed antecedents of trust from the literature. Third, the outcomes of distrust will be summarized, grouping related concepts and comparing to trust literature. If possible, a theoretical framework of distrust will be presented that incorporates the various conceptual components identified. Finally, the way evidence of distrust has been gathered will be compared to this theoretical framework. This will include tables summarizing methods for studying distrust.

### Risk of publication bias

The authors have no conflicts of interest. Ideally, additional reviewers would participate in study screening and data extraction. However, this is not feasible since this project is unfunded. We have incorporated recommendations for assessing inter-coder reliability [51] to attempt to reduce the bias introduced resulting from these limitations.

### Ethics and dissemination

This systematic review protocol has been written following the PRISMA-P guidelines and elaboration and explanation documents [47, 48]. The systematic review study is on peer-reviewed and gray literature, and thus does not include human subjects. Therefore, we will not obtain ethics clearance or documentation. We expect to publish the systematic review in a peer-reviewed journal.

## Discussion

This study investigates a potential gap in theorization and methodology within natural resource management literature. There is ongoing debate over ways to gather evidence of and conceptualize "trust" [5, 17, 43]. In contrast, it appears that the literature often assumes that the meaning of "distrust" is clear, even though at least three possible conceptualizations exist [13]. By carefully examining how evidence of distrust is gathered and the causes and consequences of this distrust, we will contribute a novel synthesis of the literature comparable to past reviews of "trust" literature [8, 57]. Any recommendations related to distrust reduction provided by these studies will also be presented.

One notable limitation of this study is the focus on clear interpretation of results as distrust. The databases and search terms in this review were carefully developed to identify articles that address the research questions. However, articles treating low trust and distrust as equivalent (i.e., where "lack of trust," "absence of trust," "untrust-," or "no trust" is mentioned but not clearly interpreted as distrust) may be missed due to the search strategy and eligibility criteria used in this review. This will lead to an underrepresentation of this conceptualization in our review–a potential concern since many authors appear to assume that low trust and distrust are equivalent. The study team felt it was essential to exclude these types of articles so that the results clearly present findings related to distrust and not just a lack of trust. We plan to keep track of the number of manuscripts excluded for this reason as a way to estimate potential under-representation in this review.

Another limitation is that some researchers may use terminology that captures constructs potentially related to distrust (e.g., bias, acceptability, legitimacy, suspicion). While relevant and related, this phrasing is beyond the scope of this review and would only be included if the authors also use one of the distrust keywords used as eligibility criteria in this study. As a result, this systematic review could be considered an initial overview of distrust in protected area and natural resource management.

## Supporting information

**S1 Table. PRISMA-P checklist.**
(DOCX)

**S1 File. Search string details for each database.**
(DOCX)

## Acknowledgments

Publication of this paper was supported, in part, by the Thomas G. Scott Publication Fund. We thank members of BE's advisory committee for input on an earlier draft of this systematic review protocol. Additionally, thanks to the Oregon State University research librarian who provided assistance in developing our search strategy.

## Author Contributions

**Conceptualization:** Brian D. Erickson, Kelly Biedenweg.

**Methodology:** Brian D. Erickson, Kelly Biedenweg.

**Supervision:** Brian D. Erickson.

**Writing – original draft:** Brian D. Erickson.

**Writing – review & editing:** Kelly Biedenweg.

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
