## [Decision Letter · Decision Letter 0]

13 Jan 2022

PONE-D-21-31426Distrust within protected area and natural resource management: A systematic review protocolPLOS ONE

Dear Dr. Erickson,

Thank you for submitting your manuscript to PLOS ONE. After careful consideration, we feel that it has merit but does not fully meet PLOS ONE’s publication criteria as it currently stands. Therefore, we invite you to submit a revised version of the manuscript that addresses the points raised during the review process. Please submit your revised manuscript by Feb 27 2022 11:59PM. If you will need more time than this to complete your revisions, please reply to this message or contact the journal office at plosone@plos.org. Please include the following items when submitting your revised manuscript:A rebuttal letter that responds to each point raised by the academic editor and reviewer(s). You should upload this letter as a separate file labeled 'Response to Reviewers'.A marked-up copy of your manuscript that highlights changes made to the original version. You should upload this as a separate file labeled 'Revised Manuscript with Track Changes'.An unmarked version of your revised paper without tracked changes. You should upload this as a separate file labeled 'Manuscript'.

We look forward to receiving your revised manuscript.

Kind regards,

Xingwei Li, Ph.D.

Academic Editor

PLOS ONE

Journal Requirements:

Additional Editor Comments:

All reviewers agreed that this manuscript must be revised. I suggest that the authors carefully revise their manuscripts based on these suggestions.

Reviewers' comments:

Reviewer #1: In this registered report protocol, Erickson and Biedenweg describe their plans for conducting a systematic review on distrust, an interesting topic with ramifications for natural resource management.

Major comments: This study would gain from a better description of the rationale leading to its conception. For example, although the article focuses on distrust, the introduction is currently more focused on trust (e.g. L28-34). Instead, I would expect a review of the existing theoretical frameworks for distrust, and their potential limitations, to see where this study fits in the literature. Additionally, there needs to be more detailed regarding the methodology; for example, in many cases, I found that the description of the data extracted from the study was unclear (see below for specific comments).

L36: specify what you mean by “polarization” In that context

L38: The statement “it is thought to both drive and result from resource conflict” requires a reference

L40-41: The concepts of “critical trust” and “effective distrust” need to be defined

L43-48: The authors explain how distrust is typically conceptualized in relation to trust; however they do not explain how trust itself is conceptualized/measured.

L84: Will the keywords be used in combination? As key strings? If so, provide exact key strings.

L94: How will the authors ensure literature saturation?

L141: By definition, a “measurement” is quantitative. Consider another terminology.

L165: Which steps will the authors take during data extraction to ensure that the study is repeatable? E.g. Will intercoder agreement be computed, will the authors go through the database multiple times, etc.

L179: what do you mean by sampling method details?

L189l,194: what do you mean by items?

L192-193: unclear what the authors mean by “distrust dimensions”

L195, 198, 202, 204, 171: who is this “reviewer”? I am confused!

L196: reliability of what?

L207: “topic analysis” might be a useful approach for this project?

L218: in this introduction, it would be nice to see a description of existing theoretical frameworks, and their potential limitations, to see where this study fits in the literature.

Table 2: what about trust* (untrustworthy, untrusting, mistrust…)

Reviewer #2: The study is an important one, as there is indeed a need to advance conceptualization, measurement, and understanding of distrust, and the research on trust is certainly more well-developed than that of distrust.

I think however the methodology could use some additional specification to ensure clarity and ability to replicate.

With regard to the primary and secondary research questions: First, those questions seem to overlap (e.g., conceptualization and measurement are mentioned in both). It would be useful to draw a clearer distinction between the two. Second, what does it mean to say "distrust conceptualized as equivalent to low/no trust," or "opposite of trust," or "distinct from trust"? It seems like distrust might be both opposite and distinct from trust. Some examples of these three conceptualizations might help anchor the meaning of each. Examples could be usefully given both for "conceptualization" and for "measurement" for each of the three categories of possibilities. Third, how do the authors distinguish between a definition and a conceptualization? Since there is a different question for definition vs. conceptualization, this seems important. Fourth, since the focus is on distrust, comparisons between recommendations for decreasing distrust vs increasing trust will be difficult (you wont be systematically sampling for trust, only distrust). This comparison may be alluded to but the methodology would not necessarily allow for a comparison within the systematic review (perhaps someone else has done a systematic review of recommendations to increase trust that could be referenced in your discussion section, but not in your results).

With regard to search strategies, Table 2 indicates it is a draft summary of keywords -- how will it be decided to add or remove keywords? The procedure indicates that no reasons will be given during the first round of screening. This seems to require elaboration -- what then will be the presumed reasons for including vs. excluding during this first round?

I'm struggling with the idea of excluding articles that assess "lack of trust" as not relevant because they do not assess "distrust." (I really appreciated the examples given in that part of the methods, to help me understand the criterion.) Given that this is one of the 3 already-identified conceptualizations of distrust, it seems like those articles should be included. On the other hand, I suppose the difficulty is that then the researchers might then also need to include all articles about "trust" or measuring "trust" because all such articles would surely also have something to say about "low trust"/lack of trust. I'd recommend that the authors track and specify how many articles are screened out specifically for this reason (as opposed to other exclusion reasons) so that they might better assess the risk and extent of "under-representation" (which they do note as a risk) due to the application of this methodological choice.

Other comments:

Note that I think some would argue that trust is not "robustly conceptualized and measured" (e.g., see McEvily, B., & Tortoriello, M. (2011). Measuring trust in organisational research: Review and recommendations. Journal of Trust Research, 1(1), 23-63.). Nonetheless, it is more well-conceptualized and measured than is distrust. Or, at least, more attention has been given to date.

No citations are given for the statements in lines 53-56 even though it states "multiple studies...".

Reviewer #3: This is a compelling registration. I think the authors are generous in saying that trust is robustly conceptualized and measured (many authors fail to define it, in my reading), but they are spot on in identifying that distrust is so underdefined. This will be a needed literature review, and I especially find the authors' attenton to how researchers fail to distinguish low trust, the opposite of trust, and a different concept from trust, and to whether the actions recommended to reduce distrust are similar or different from those recommended to build trust, compelling.

With respect to methods, the authors have carefully thought out their approach, and I agreed with the vast majority of their decisions: the use of Krippendorf's alpha and cutoff, assessment criteria for what counts as NRM, using reviewers not blind to the hypotheses because of funding considerations, and conducting largely descriptive "analyses." I had one concern about their exclusion of articles that don't explicitly discuss distrust more than a brief mention -- they may miss much about the antecedents and consequences of distrust, things they wish to characterize, if they exclude articles that only mention distrust in passing -- but I found their defense of this decision (in the Discussion) reasonable.

My feedback is minor:

Lines 58-66: There is either a subtle difference or a redundancy between the primary and secondary questions, so let me flag it and check the logic. (1) Subtle difference: How is distrust conceptualized, period; if it is measured, how is it conceptualized -- is conceptualization different for this subset of cases than the whole? (2) Redundancy: The question "how is distrust conceptualized" appears twice.

The authors make a compelling case that distrust is often measured by accident. Perhaps the useful thing to do here would be to separate the primary question into two -- (1) how is distrust conceptualized broadly? (2) how is it measured? -- and add two sub-questions to the secondary questions -- (1) when distrust is accidentally measured, how is it conceptualized (presumably posthoc)? [likely this will appear in the discussion sections of relevant papers] (2) when distrust is accidentally measured, what are the measurements that pick it up?

Lines 63-66: This is unclear to me, and it might just be because the authors didn't unpack it: what is the difference between conceptualize and define? I could see one as being the theoretical motivation and the other as being how the definition is written down (a generous assumption that it is written down, as most papers don't even define what they mean by trust).

Lines 92-93: Any reason for the choice of 1990 as the start year? I see the CIFOR's earliest year is 1993. I also notice from the S1 File that one reason for using NOFT for ProQuest is to keep the search comparable to the search in WoS. (I agree with the choice to use NOFT based on the second reason provided: avoiding lots of extraneous hits.) If the goal of the review is to characterize the existing literature, why attempt to make sampling across databases comparable by omitting anything earlier than 1990 and going for comparability in searches across databases?

Lines 170-204 (on data extraction):

- The authors might consider making things easier for themselves by coding with overarching categories as they go. For example, along with the Trust Definition and Distrust Definition columns, consider a Distrust Type column with levels like Equivalent, Opposite, Distinct; since the authors expect these to be common usages, tagging this as they go through may streamline the process. Similarly, a separate columns for coding might be helpful for Distrustee Type (e.g., government, NGO, stakeholders)

- Many papers may fail to define trust or distrust: what about putting the context of usage in these columns, if a definition isn't available?

- Maybe put Action and Subject in different columns for easy sorting (e.g., in Action: management, conservation, restoration; in Subject: fisheries, forests, air, species)

- I recommend another column to go with How Developed: if the study adapted or replicated measures, what was the source (e.g., the citation they provide for those measures); it will make it easier to track them down later if needed (and to potentially sort out the source from the dataset if needed, e.g., to remove "duplicates")

---

## [Author Response · Author response to Decision Letter 0]

28 Jan 2022

PONE-D-21-31426

Distrust within protected area and natural resource management: A systematic review protocol

PLOS ONE

Dear Dr. Erickson,

Thank you for submitting your manuscript to PLOS ONE. After careful consideration, we feel that it has merit but does not fully meet PLOS ONE’s publication criteria as it currently stands. Therefore, we invite you to submit a revised version of the manuscript that addresses the points raised during the review process.

We look forward to receiving your revised manuscript.

Kind regards,

Xingwei Li, Ph.D.

Academic Editor

PLOS ONE

Journal Requirements:

We adjusted file names, removed the secondary author’s email, and modified capitalization in one heading. Beyond that, we did not identify additional style requirement changes needed. 

Our ethics and dissemination statement is within the Methods section just before the Discussion. This seems to match the guidelines, but we are happy to make adjustments, if needed.

Additional Editor Comments:

All reviewers agreed that this manuscript must be revised. I suggest that the authors carefully revise their manuscripts based on these suggestions.

Thank you for obtaining the extremely constructive reviews of our manuscript. We have undertaken a major revision to address the suggestions and concerns, which has greatly improved the manuscript. Among the three reviews we identified two main concerns, which we addressed as follows:

1. Underdeveloped or unclear methods, especially around data extraction and research questions. We have modified the research questions to better capture our focus and intent, and we had added clarity around how we will approach this systematic review in hopes of increasing replicability.

2. Underdeveloped connections to trust and distrust theory. Addressed by adding a theoretical background section that introduces how trust and distrust are often treated, with an emphasis on the natural resource management literature when possible.

Other changes include addition of citations and modification of the abstract. Below we provide a point-by-point response to all reviewer comments, in which we indicate all changes made to the manuscript.

Reviewers' comments:

Reviewer #1: In this registered report protocol, Erickson and Biedenweg describe their plans for conducting a systematic review on distrust, an interesting topic with ramifications for natural resource management.

Major comments: This study would gain from a better description of the rationale leading to its conception. For example, although the article focuses on distrust, the introduction is currently more focused on trust (e.g. L28-34). Instead, I would expect a review of the existing theoretical frameworks for distrust, and their potential limitations, to see where this study fits in the literature. Additionally, there needs to be more detailed regarding the methodology; for example, in many cases, I found that the description of the data extracted from the study was unclear (see below for specific comments).

We thank the reviewer for their time and effort to review our manuscript and for their attention to detail. In an effort to keep the protocol concise, we initially limited theoretical development of trust and distrust. Your comment highlighted that this led to underdevelopment of our rationale for the review. We’ve added in a theoretical background section. We have also made adjustments to the methods to increase clarity.

L36: specify what you mean by “polarization” In that context

We understand how this term carries multiple meanings. To maintain the focus in the paragraph, we removed this term instead of defining and elaborating.

L38: The statement “it is thought to both drive and result from resource conflict” requires a reference

Inserted a citation to Nie (2003).

L40-41: The concepts of “critical trust” and “effective distrust” need to be defined

We’ve added explanations of both concepts. 

L43-48: The authors explain how distrust is typically conceptualized in relation to trust; however they do not explain how trust itself is conceptualized/measured.

We have added a section on how trust is conceptualized and intersperse discussion of measurement throughout the introduction/background.

L84: Will the keywords be used in combination? As key strings? If so, provide exact key strings.

We attempted to clarify this by noting the keywords used were shown in Table 2 and that we were using database-specific Boolean search strategies, shown in S1 File.

L94: How will the authors ensure literature saturation?

Great recommendation. We added in a sentence toward the end of our article screening section explaining that we plan to manually search the reference lists of the articles included in the review

L141: By definition, a “measurement” is quantitative. Consider another terminology.

This comment was challenging for us, but we ultimately agree with you. We have updated the manuscript to focus on “gathering evidence of distrust.” In our minds, this includes measuring distrust (quantitative) as well as qualitative data collection approaches. 

L165: Which steps will the authors take during data extraction to ensure that the study is repeatable? E.g. Will intercoder agreement be computed, will the authors go through the database multiple times, etc.

 We added a section explaining this. Similar to the screening processes, a second coder will extract data from a subset of articles being reviewed (10%). Intercoder reliability will be assessed and disagreements will be discussed.

L179: what do you mean by sampling method details?

This was meant as a category to capture additional notes and comments related to the methods. We modified this to clarify it is for “comments on methods.”

L189l,194: what do you mean by items?

We removed this phrase. “Items” was meant to refer to each question on a questionnaire that might be used to measure distrust. For clarify, we simplified this to “Questions asked”

L192-193: unclear what the authors mean by “distrust dimensions”

Trust theorists debate whether trust is unidimensional, bidimensional, or multidimensional. Some debated dimensions include ability, benevolence, integrity, reliability as well as different types of trust (rational, affinity, procedural, dispositional). This bullet was meant to provide space for any author-identified dimensions of distrust being studied. Since we only know of one paper that identifies multiple possible components of distrust a priori (Pytlik Zillig et al. 2016), we will plan to capture author-identified dimensions elsewhere in the data extraction form. 

L195, 198, 202, 204, 171: who is this “reviewer”? I am confused!

We removed the word “reviewer.” This was meant to add clarity that the data extraction form was providing a space for the person doing the data extraction (specifically BE, the lead author of this protocol) to provide comments on different elements of the data extraction form.

L196: reliability of what?

This was intended to capture any text the manuscript authors might have included related to the reliability of any indices created to capture a construct (i.e., distrust).

L207: “topic analysis” might be a useful approach for this project?

We appreciate the recommendation from the reviewer. Our understanding of topic analysis is that it is a quantitative approach that uses machine learning to identify major themes in a corpus. While the approach sounds promising, we believe a thematic coding approach will better enable us to connect discussions to existing theorization around trust and distrust. 

L218: in this introduction, it would be nice to see a description of existing theoretical frameworks, and their potential limitations, to see where this study fits in the literature.

We have updated the introduction to include an overview of some theoretical frameworks related to trust and distrust. We have tried to balance conciseness and completeness.

Table 2: what about trust* (untrustworthy, untrusting, mistrust…)

We added an additional keyword to our search, “untrust*”, which should catch both “untrustworthy” and “untrusting.” We also added astericks to our keyword “no trust” to broaden it to “no* trust*”.

 

Reviewer #2: The study is an important one, as there is indeed a need to advance conceptualization, measurement, and understanding of distrust, and the research on trust is certainly more well-developed than that of distrust.

Thank you for the encouragement and critical eye in your review! 

I think however the methodology could use some additional specification to ensure clarity and ability to replicate.

With regard to the primary and secondary research questions: First, those questions seem to overlap (e.g., conceptualization and measurement are mentioned in both). It would be useful to draw a clearer distinction between the two. 

We agree that the questions overlapped. We simplified the primary question and split it into two questions. The secondary questions were edited to remove redundancy and add clarity.

Second, what does it mean to say "distrust conceptualized as equivalent to low/no trust," or "opposite of trust," or "distinct from trust"? It seems like distrust might be both opposite and distinct from trust. Some examples of these three conceptualizations might help anchor the meaning of each. Examples could be usefully given both for "conceptualization" and for "measurement" for each of the three categories of possibilities. 

We have removed this question as it was a restatement of the broader question, “how is distrust conceptualized?” In addition, we have added examples and explanation in a theoretical background section. 

Third, how do the authors distinguish between a definition and a conceptualization? Since there is a different question for definition vs. conceptualization, this seems important. 

We view definition as a subset of conceptualization. Based on what we have seen, we expect few authors will explicitly provide (i.e., write) a definition of distrust. If they do provide one, we will record it. Otherwise, we will try to determine, based on language use, which conceptualization (equivalent, opposite, or distinct) they are using. 

Fourth, since the focus is on distrust, comparisons between recommendations for decreasing distrust vs increasing trust will be difficult (you wont be systematically sampling for trust, only distrust). This comparison may be alluded to but the methodology would not necessarily allow for a comparison within the systematic review (perhaps someone else has done a systematic review of recommendations to increase trust that could be referenced in your discussion section, but not in your results).

This is a great point! We had not considered that our sampling approach would not allow for the systematic comparison needed to answer this question. We removed the question about comparison between approaches to building trust and reducing distrust. We will report any recommendations for reducing distrust found in the literature; however, our initial impression is that explicit recommendations around distrust reduction are limited or non-existent. The discussion, as recommended, will likely note that ways to “build trust” may provide insight but were beyond the scope of this review.

With regard to search strategies, Table 2 indicates it is a draft summary of keywords -- how will it be decided to add or remove keywords? 

The word “draft” was included based on item 10 in Shamseer et al. (2015), which reads “Present draft of search strategy to be used for at least one electronic database, including planned limits, such that it could be repeated” (p. 9). For clarity, “draft” was removed as the keyword list is not intended to change once the review begins.

The procedure indicates that no reasons will be given during the first round of screening. This seems to require elaboration -- what then will be the presumed reasons for including vs. excluding during this first round?

We see that our writing was not clear here. This was meant to say that the lead review author would screen titles, abstracts, and keywords yielded by the search against the inclusion criteria. This line was meant to say that detailed notes about why a study was coded as “include,” “exclude,” or “unsure” would not be made at this stage due to the anticipated large number of studies to screen. For clarity, the sentence in question was deleted.

I'm struggling with the idea of excluding articles that assess "lack of trust" as not relevant because they do not assess "distrust." (I really appreciated the examples given in that part of the methods, to help me understand the criterion.) Given that this is one of the 3 already-identified conceptualizations of distrust, it seems like those articles should be included. On the other hand, I suppose the difficulty is that then the researchers might then also need to include all articles about "trust" or measuring "trust" because all such articles would surely also have something to say about "low trust"/lack of trust. I'd recommend that the authors track and specify how many articles are screened out specifically for this reason (as opposed to other exclusion reasons) so that they might better assess the risk and extent of "under-representation" (which they do note as a risk) due to the application of this methodological choice.

This was a challenge for us as well! This challenge is part of the motivation for the review since most articles we have read presume a lack of trust/low trust IS the same as distrust. We appreciate the recommendation to track articles excluded for this reason as a way to track potential under-representation. We added this to the “eligible measures” section. 

We also decided that in the first round of screening, articles mentioning any distrust keywords would be passed to round 2. During round 2, these articles would be assessed for how lack of trust is interpreted (distrust or just trust?).

Other comments:

Note that I think some would argue that trust is not "robustly conceptualized and measured" (e.g., see McEvily, B., & Tortoriello, M. (2011). Measuring trust in organisational research: Review and recommendations. Journal of Trust Research, 1(1), 23-63.). Nonetheless, it is more well-conceptualized and measured than is distrust. Or, at least, more attention has been given to date.

We modified our wording by removing the term “robust” and acknowledging the more commonly occurring theory and emphasis on measurement of trust.

No citations are given for the statements in lines 53-56 even though it states "multiple studies...".

We added a few citations to support this statement.

 

Reviewer #3: This is a compelling registration. I think the authors are generous in saying that trust is robustly conceptualized and measured (many authors fail to define it, in my reading), but they are spot on in identifying that distrust is so underdefined. 

Thanks for noticing this. We realized we were making a relative statement, in comparison to distrust in the literature, but that this came across as though trust was well-developed. We slightly modified our wording.

This will be a needed literature review, and I especially find the authors' attention to how researchers fail to distinguish low trust, the opposite of trust, and a different concept from trust, and to whether the actions recommended to reduce distrust are similar or different from those recommended to build trust, compelling.

Thank you for your encouragement and attention to detail in your recommendations. 

While we are interested in the comparison between recommendations for trust building and distrust reduction as well, Reviewer #2 pointed out that the selection methods do not adequately sample to answer this question since studies only focusing on trust were excluded. While we intended to report on distrust reduction methods and may hint at recommendations for building trust in the Discussion, the review will no longer explicitly try to compare these.

With respect to methods, the authors have carefully thought out their approach, and I agreed with the vast majority of their decisions: the use of Krippendorf's alpha and cutoff, assessment criteria for what counts as NRM, using reviewers not blind to the hypotheses because of funding considerations, and conducting largely descriptive "analyses." I had one concern about their exclusion of articles that don't explicitly discuss distrust more than a brief mention -- they may miss much about the antecedents and consequences of distrust, things they wish to characterize, if they exclude articles that only mention distrust in passing -- but I found their defense of this decision (in the Discussion) reasonable.

We agree with your concern and will keep it in mind as we consider the scope and implications of our review. This challenge is at the heart of this review: if we are not clear about what distrust is or how to measure it, we cannot be sure whether findings of low trust tell us about distrust or not.

My feedback is minor:

Lines 58-66: There is either a subtle difference or a redundancy between the primary and secondary questions, so let me flag it and check the logic. (1) Subtle difference: How is distrust conceptualized, period; if it is measured, how is it conceptualized -- is conceptualization different for this subset of cases than the whole? (2) Redundancy: The question "how is distrust conceptualized" appears twice.

The authors make a compelling case that distrust is often measured by accident. Perhaps the useful thing to do here would be to separate the primary question into two -- (1) how is distrust conceptualized broadly? (2) how is it measured? -- and add two sub-questions to the secondary questions -- (1) when distrust is accidentally measured, how is it conceptualized (presumably posthoc)? [likely this will appear in the discussion sections of relevant papers] (2) when distrust is accidentally measured, what are the measurements that pick it up?

We appreciate the careful eye. We did not intend to imply that conceptualization would differ in studies that do and do not define distrust. Instead, the questions were redundant. The original intention was to connect the primary and secondary questions. We have taken the second recommendation: divide the primary question into two and add two secondary research questions regarding distrust emerging from research (i.e., accidentally measuring it).

Lines 63-66: This is unclear to me, and it might just be because the authors didn't unpack it: what is the difference between conceptualize and define? I could see one as being the theoretical motivation and the other as being how the definition is written down (a generous assumption that it is written down, as most papers don't even define what they mean by trust).

We intended “conceptualize” to be an overarching term. “Define” was specifically meant to capture how a definition of distrust is written down, if at all. We have removed the redundancy and clarified our questions to hopefully avoid our lack of clarity.

Lines 92-93: Any reason for the choice of 1990 as the start year? I see the CIFOR's earliest year is 1993. I also notice from the S1 File that one reason for using NOFT for ProQuest is to keep the search comparable to the search in WoS. (I agree with the choice to use NOFT based on the second reason provided: avoiding lots of extraneous hits.) If the goal of the review is to characterize the existing literature, why attempt to make sampling across databases comparable by omitting anything earlier than 1990 and going for comparability in searches across databases?

Our initial goal was to identify "current” conceptualizations of distrust, which limited the search to 2010-Present. Since several highly sighted papers on trust in NRM appeared in the late 1990s and early 2000s, we decided to expand the search to 1990-Present. Upon reflection, the 1990 cut off feels artificial and we decided to remove any time restrictions.

Lines 170-204 (on data extraction):

- The authors might consider making things easier for themselves by coding with overarching categories as they go. For example, along with the Trust Definition and Distrust Definition columns, consider a Distrust Type column with levels like Equivalent, Opposite, Distinct; since the authors expect these to be common usages, tagging this as they go through may streamline the process. Similarly, a separate columns for coding might be helpful for Distrustee Type (e.g., government, NGO, stakeholders)

This is a great suggestion. We were trying to indicate this by the “Distrust Conceptualization” bullet. We clarified in the parentheses that we would code as equivalent, opposite, distinct, or unclear. The recommendation of “distrustee type” is well-intended, but we are unsure how to implement it. For example, some studies consider NGOs as stakeholders while others place NGOs and stakeholders in separate categories. Some sample “local communities” while others focus specifically on resource users, community members with specific roles, etc. During data extraction we plan to use the term for the population sampled in the study. Ideally, we will be able to simplify this during analysis but we are not comfortable deciding a head of time what these categories are.

- Many papers may fail to define trust or distrust: what about putting the context of usage in these columns, if a definition isn't available?

We added a bullet for sample text to provide a space to show how trust and distrust are used within the manuscript being reviewed.

- Maybe put Action and Subject in different columns for easy sorting (e.g., in Action: management, conservation, restoration; in Subject: fisheries, forests, air, species)

While the “action” is critical to understanding trust (i.e., party A distrust party B to do action X), an initial piloting of our data extraction form showed that it is very difficult to identify the action being assessed. First, most manuscripts we have seen do not specify what questions were used to specifically measure distrust. For example, many qualitative studies describe distrust as a theme that emerged from the research, but the description of themes included in interview guides does not include (dis)trust as a topic of discussion. We plan to analyze the actions being measured through a look at the specific questions used to measure distrust.

- I recommend another column to go with How Developed: if the study adapted or replicated measures, what was the source (e.g., the citation they provide for those measures); it will make it easier to track them down later if needed (and to potentially sort out the source from the dataset if needed, e.g., to remove "duplicates")

Great idea! We have added this column.

---

## [Decision Letter · Decision Letter 1]

1 Mar 2022

Distrust within protected area and natural resource management: A systematic review protocol

PONE-D-21-31426R1

Dear Dr. Erickson,

We’re pleased to inform you that your manuscript has been judged scientifically suitable for publication and will be formally accepted for publication once it meets all outstanding technical requirements.

Kind regards,

Xingwei Li, Ph.D.

Academic Editor

PLOS ONE

Additional Editor Comments (optional):

I agree with the reviewers' suggestion to accept as the current version.

Reviewers' comments:

Reviewer's Responses to Questions

6. Review Comments to the Author

Reviewer #1: The authors did a great job at addressing previous comments; I am looking forward to reading their findings!

Reviewer #3: I reviewed this report on its previous submission, and it's clear the authors took the time to think hard about each of the reviewers' suggestions, accepting most of them. The study will be stronger for it (and ideally, the theoretical background the authors wrote for us can be easily converted into an introduction for the eventual manuscript). I look forward to seeing the authors' results!

Just a quick note that I could not find precise details about which repository the authors will use to release their data publicly, only a statement that they will do so. Perhaps they can clarify directly to the editor which repository they plan to use.

---

## [Editor Report · Acceptance letter]

7 Mar 2022

PONE-D-21-31426R1 

Distrust within protected area and natural resource management: A systematic review protocol 

Dear Dr. Erickson:

I'm pleased to inform you that your manuscript has been deemed suitable for publication in PLOS ONE. Congratulations! Your manuscript is now with our production department. 

Kind regards, 

on behalf of

Prof. Dr. Xingwei Li 

Academic Editor

PLOS ONE